# An antibonding valence band maximum enables defect-tolerant and stable GeSe photovoltaics

Shun-Chang Liu [1,2,6], Chen-Min Dai[3,6], Yimeng Min[4], Yi Hou [4], Andrew H. Proppe [4], Ying Zhou [5], Chao Chen[5], Shiyou Chen [3], Jiang Tang [5], Ding-Jiang Xue [1,2✉], Edward H. Sargent [4✉] & Jin-Song Hu [1,2✉]

In lead–halide perovskites, antibonding states at the valence band maximum (VBM)—the result of Pb 6$s$-I 5$p$ coupling—enable defect-tolerant properties; however, questions surrounding stability, and a reliance on lead, remain challenges for perovskite solar cells. Here, we report that binary GeSe has a perovskite-like antibonding VBM arising from Ge 4$s$-Se 4$p$ coupling; and that it exhibits similarly shallow bulk defects combined with high stability. We find that the deep defect density in bulk GeSe is ~10$^{12}$ cm$^{-3}$. We devise therefore a surface passivation strategy, and find that the resulting GeSe solar cells achieve a certified power conversion efficiency of 5.2%, 3.7 times higher than the best previously-reported GeSe photovoltaics. Unencapsulated devices show no efficiency loss after 12 months of storage in ambient conditions; 1100 hours under maximum power point tracking; a total ultraviolet irradiation dosage of 15 kWh m$^{-2}$; and 60 thermal cycles from −40 to 85 °C.

---

[1] Beijing National Laboratory for Molecular Sciences (BNLMS), CAS Key Laboratory of Molecular Nanostructure and Nanotechnology, Institute of Chemistry, Chinese Academy of Sciences, Beijing 100190, China. [2] University of Chinese Academy of Sciences, Beijing 100049, China. [3] Key Laboratory of Polar Materials and Devices (MOE), East China Normal University, Shanghai 200241, China. [4] Department of Electrical and Computer Engineering, University of Toronto, Toronto Ontario M5S 1A4, Canada. [5] Wuhan National Laboratory for Optoelectronics (WNLO), Huazhong University of Science and Technology, Wuhan 430074, China. [6] These authors contributed equally: Shun-Chang Liu, Chen-Min Dai. ✉email: djxue@iccas.ac.cn; ted.sargent@utoronto.ca; hujs@iccas.ac.cn

The macroscopic properties of a semiconductor depend on the chemical bonding between its constituent elements: electrostatics for ionic systems, and orbital hybridization for covalent systems. In an organic–inorganic hybrid perovskite solar cells (PSCs), power conversion efficiencies (PCEs) have increased from an initial 3.8%[1] to a certified 25.2%[2] in the last decade. The electronic structure of perovskites—Pb $6s$–I $5p$ antibonding states at the valence band maximum (VBM), contrasting with semiconductors such as GaAs and GaN that have a bonding VBM—is a key to the extraordinary performance of perovskite-based optoelectronics[3–10]. Because of this antibonding feature, defects in perovskites are confined to shallow states close to the band edges, instead of introducing states within the bandgap[11–16].

Another bonding feature in perovskites is high ionicity, which enables thin-film fabrication, wherein perovskites form highly crystalline materials when prepared even at room temperature[17,18]. However, this high ionicity makes perovskites easily soluble in water and sensitive to moisture, inducing performance degradation[19,20]. Ionic perovskites also exhibit ion migration, a reason for $J$-$V$ hysteresis[21–23].

We reasoned that the combination of strong covalent bonding, coupled with high stability and a perovskite-like antibonding VBM electronic structure, could potentially add new defect-tolerant materials to photovoltaics.

We turned our attention to germanium monoselenide (GeSe), a material whose similar Pauling electronegativity of Ge (2.01) and Se (2.55) suggests a covalent semiconductor. GeSe has recently emerged as a promising absorber material for photovoltaics owing to its suitable bandgap (~1.14 eV), high absorption coefficient (greater than $10^5$ cm$^{-1}$), high carrier mobility (~128 cm$^2$ V$^{-1}$ s$^{-1}$), and its earth-abundant, and Pb-free composition[24–31]. Its sublimation characteristic enables in-situ self purification of the raw material, leaving impurities in the sublimation source during film deposition[24]. Since GeSe (s) sublimes to GeSe (g) without decomposing into elemental species, it avoids the formation of undesired Ge and Se interstitials[30]. Whether or not GeSe can exhibit defect tolerance analogous to that of Pb-based perovskites, especially in light of its perovskite-like $ns^2$ electronic configuration, has so far remained unclear.

We began by investigating defects in GeSe. We find that GeSe, which has a $4s^2$ electronic configuration, possesses a perovskite-like antibonding VBM arising from Ge $4s$–Se $4p$ coupling; and that this leads to shallow bulk defects, and it also prevents GeSe oxidation. We then find that surface defects in GeSe photovoltaics have played a major role in device performance until now, and we, therefore, develop surface-passivated GeSe solar cells. These achieve a certified PCE of 5.2%, surpassing the best previously-reported GeSe results 3.7 fold. These devices exhibit excellent stability as required of thin-film photovoltaic modules (IEC 61646).

## Results

**Antibonding coupling for defect-tolerant GeSe**. GeSe crystallizes in an orthorhombic layered structure with the *Pnma* 62 space group (Fig. 1a). Both Ge and Se atoms are three fold coordinated with each other. There is only one type of Ge and Se: GeSe is a binary chalcogenide both chemically and structurally. There are therefore only six possible point defects in GeSe: cation vacancy ($V_{Ge}$), anion vacancy ($V_{Se}$), cation interstitial ($Ge_i$), anion interstitial ($Se_i$), cation-replace-anion antisite ($Ge_{Se}$), and anion-replace-cation antisite ($Se_{Ge}$). This is simpler than in multicomponent semiconductors such as Cu(In,Ga)Se$_2$ (CIGS) and Cu$_2$ZnSn(S,Se)$_4$ (CZTSSe)[32,33].

We first used density functional theory (DFT) to calculate the bandstructure, density of states (DOS), and partial DOS of GeSe, since the electronic properties of point defects depend sensitively on the structure. The conduction band minimum (CBM) of GeSe is dominated by the Ge $4p$ orbital, with significant coupling with the Se $4p$ orbital; and negligible coupling with the Se $4s$ orbital (Fig. 1b). This indicates the strong covalent character of GeSe, agreeing well with the above electronegativity analysis, and differing from perovskites with their high ionicity. As for the VBM, it is predominantly made up of the Se $4p$ orbital and the Ge $4p$ orbital due to $p$–$p$ coupling, with a substantial contribution from the Ge $4s$ orbital. This is seen in the bandstructure of GeSe and the partial DOS of the Ge $4s$ orbital (Fig. 1b).

The reason that the inner-shell Ge $4s$ orbital is present in the VBM is illustrated through an atomic orbital picture (Fig. 1c): the Ge $4s$ and $4p$ orbitals are too far apart in energy to hybridize directly[4,34]; and the Se $4p$ orbital is close to the Ge $4s$ orbital in energy, allowing these to the couple and giving rise to a filled antibonding orbital at the VBM[3,4]. The CBM also has an antibonding character originating from the Ge $4p$–Se $4p$ coupling. In the lone pair model, the asymmetrically layered-crystal structure of GeSe arising from the stereochemically active lone pairs accounts for the contribution of the Ge $4s$ orbital to the VBM[4]. This differs from other IV–VI materials such as PbS and SnTe, which have symmetric structures[4,7]. The partial oxidation of Ge to its Ge$^{2+}$ oxidation state contributes an antibonding $4s$ character to the VBM, as in lead–halide perovskites.

We then calculated the formation energies and transition energy levels of the six possible point defects in GeSe mentioned above; we used the generalized gradient approximation (GGA) in these studies. The most striking observation is the high formation energies for all the defects, higher than 1.2 eV in their neutral charge states (Fig. 1e, f). This is in contrast with CH$_3$NH$_3$PbI$_3$, in which defects have low formation energies (close to zero)[14,15]. This is attributed to the stronger covalent Ge–Se bonds compared with the soft Pb–I bonds in perovskites. The second notable feature is that $V_{Ge}$, with the lowest formation energy, has a shallow level with (-/0) and (2-/-) transition energy levels only 0.05 and 0.15 eV above the VBM, whereas defects with deep levels such as Ge$_{Se}$, Ge$_i$, $V_{Se}$, and Se$_{Ge}$ have high formation energies (Supplementary Fig. 1). These are reconfirmed by Heyd-Scuseria-Ernzerhof (HSE) calculations (Supplementary Fig. 2).

The low formation energy of $V_{Ge}$ is attributed to energetically unfavorable Ge $4s$–Se $4p$ antibonding coupling, where the fully occupied antibonding state has no electronic energy[14]. The shallow nature of $V_{Ge}$ originates from the antibonding state at the VBM, a defect-tolerant electronic structure known to lead to shallow defects (Supplementary Fig. 3). This state pushes the VBM to a higher level such that the acceptor defect is shallower near the VBM than that without strong $s$–$p$ coupling. The strongly covalent GeSe with an antibonding VBM, therefore, exhibits mostly shallow defects.

We used deep-level transient spectroscopy (DLTS) to investigate the defect energy levels, concentrations, and types in semiconductor devices. The DLTS spectrum of GeSe photovoltaic devices fabricated using the previously-reported rapid thermal sublimation approach[24] is shown in Supplementary Fig. 4. Two positive peaks denoted as H1 and H2 are observed at 285 K and 310 K, indicating two types of hole trap defects in the GeSe film. The activation energy ($E_a$) and capture cross-section ($\sigma$) values determined from the Arrhenius plots are 0.35 eV and $4.3 \times 10^{-23}$ cm$^2$ in H1, and 0.51 eV and $7.6 \times 10^{-21}$ cm$^2$ in H2, respectively (Fig. 1d). The concentration of defects ($N_T$) calculated from the equation of $N_T = 2\Delta C^* N_A / C_0$ ($N_A$ is the net acceptor concentration in GeSe film) are $1.3 \times 10^{12}$ cm$^{-3}$ for H1 and $3.0 \times 10^{12}$ cm$^{-3}$ for H2, lower

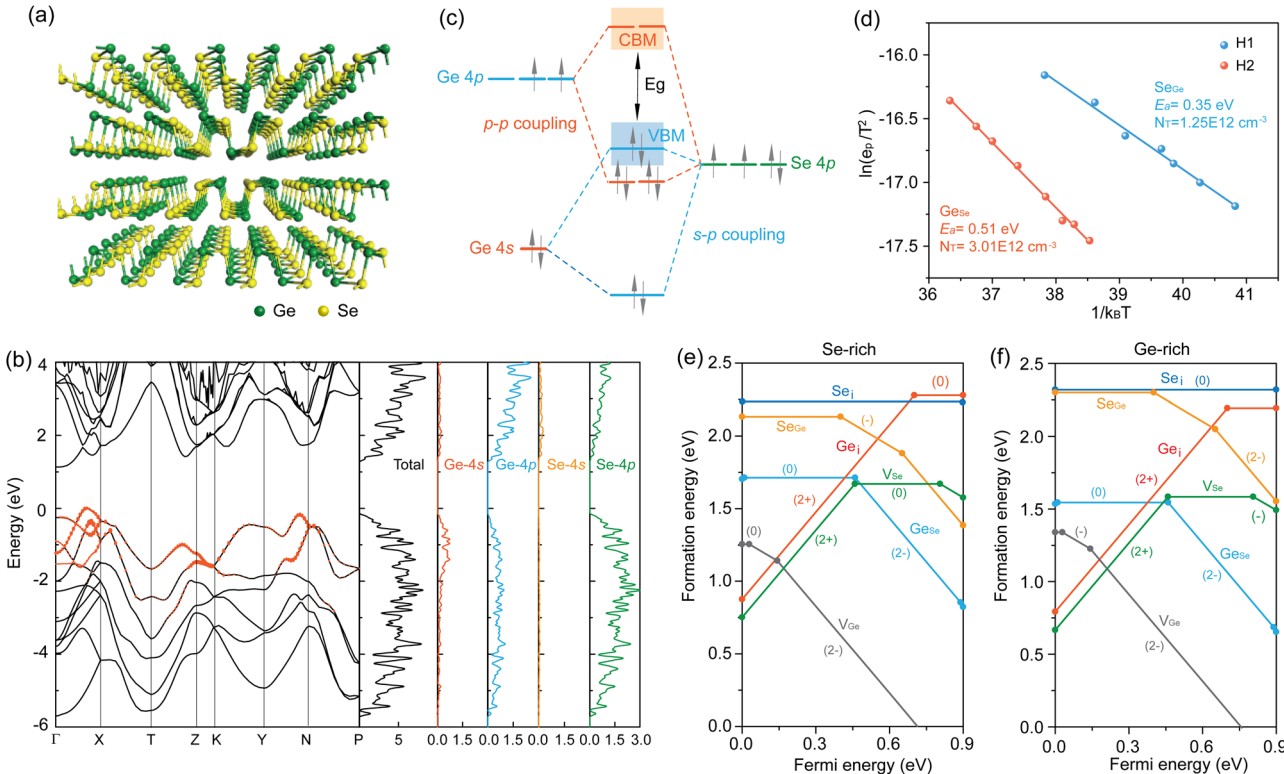

**Fig. 1 Point defect properties of GeSe. a** Crystal structure of GeSe. **b** Calculated bandstructure, density of states (DOS), and partial DOS projected on different elements of GeSe. **c** Schematic energy level diagram of the interactions in GeSe. **d** Arrhenius plots obtained from DLTS experiments. The solid lines represent the best fits to experimental data. Calculated formation energies of intrinsic point defects in GeSe under **e** Se-rich and **f** Ge-rich conditions as a function of the Fermi energy.

than state-of-art chalcogenides such as CIGS ($\sim 4.2 \times 10^{13}$ cm$^{-3}$) and CZTS ($\sim 3.7 \times 10^{14}$ cm$^{-3}$)[35,36].

There are only two deep acceptor defects, $Se_{Ge}$ and $Ge_{Se}$. We associate the H1 and H2 defects observed in the DLTS fitting results at 0.35 eV and 0.51 eV with $Se_{Ge}$ and $Ge_{Se}$, respectively. No $V_{Ge}$ is observed in the DLTS measurement since $V_{Ge}$ is too shallow to produce a response in the DLTS signal. Note that the densities of deep defects in GeSe including $Se_{Ge}$ and $Ge_{Se}$ are at a magnitude of $10^{12}$ cm$^{-3}$, well below the bulk density of GeSe ($\sim 10^{15}$) cm$^{-3}$ dominated by $V_{Ge}$[24,30]. Admittance spectroscopy (AS) measurements further confirmed the low densities of deep-level defects in GeSe. The $E_a$ values deduced from the Arrhenius plots are 0.29 eV and 0.45 eV, while the integrated defect densities of these two defects are $1.6 \times 10^{13}$ cm$^{-3}$ and $3.5 \times 10^{12}$ cm$^{-3}$ (Supplementary Fig. 5), respectively. In sum, defects with low formation energies generate only shallow levels, whereas deep-level defects have high formation energies and their density is low.

**Photovoltaic device performance**. When we fabricated devices using an architecture of ITO/CdS/GeSe/Au, we obtained a low PCE of 1.4%, with a $V_{oc}$ of 0.23 V, a $J_{sc}$ of 15.7 mA cm$^{-2}$, and a FF of 40% (Fig. 2a). We reasoned that this inferior performance could arise from surface states. We then characterized the density of interfacial defects at the CdS/GeSe heterojunction through a combination of capacitance-voltage (C-V) profiling and drive-level capacitance profiling (DLCP) measurements. C-V measurements are sensitive to free carriers as well as bulk and interfacial defects, while DLCP measurements are responsive to free carriers and bulk defects[37,38]. Thus, the density of interfacial defects at the heterojunction is estimated by subtracting $N_{DLCP}$

(defect density calculated from DLCP) from $N_{C-V}$ (defect density calculated from C-V). We calculated an interfacial defect density of $2 \times 10^{12}$ cm$^{-2}$ at the GeSe/CdS interface (Fig. 2b), which can lead to severe recombination losses.

We focused therefore on surface passivation of GeSe films. We posited that $Sb_2Se_3$ would act as a bridge between CdS and GeSe. Recently, a buried CdS/$Sb_2Se_3$ homojunction has been reported to arise due to the interfacial diffusion of cadmium, forming a good interface between CdS and $Sb_2Se_3$ layers[39]. We applied DFT to investigate the interface formation energy between GeSe and $Sb_2Se_3$ with a preferred orientation of [111] for GeSe and [221] for $Sb_2Se_3$. The formation energy is $-0.12$ eV, indicating that the growth of GeSe on $Sb_2Se_3$ is feasible. X-ray diffraction (XRD) was then used to characterize the orientation of both GeSe and $Sb_2Se_3$ layers. When we deposited a GeSe film onto a [211]-oriented $Sb_2Se_3$ layer (Supplementary Fig. 6), we found that the modified GeSe has a preferred [111] orientation, whereas the peaks of (200) and (400) with the lowest surface energies for GeSe disappear completely (Supplementary Fig. 7). This confirms the strong interaction between [211]-oriented $Sb_2Se_3$ and [111]-oriented GeSe, in agreement with theoretical calculations.

Photovoltaic devices that use the modified GeSe films are improved with a $V_{oc}$ of 0.36 V, a $J_{sc}$ of 26.9 mA cm$^{-2}$, a FF of 54%, and a PCE of 5.2% (Fig. 2a). This efficiency is 3.7× higher than that of control devices. DFT calculations were used to study further the role of $Sb_2Se_3$: dangling bonds on the surface of [111]-oriented GeSe film lead to localized states inside the bandgap, causing recombination losses (Fig. 2c), whereas the electron distribution becomes delocalized following modification with $Sb_2Se_3$ (Fig. 2d). C-V profiling and DLCP measurements reveal an order of magnitude lower interfacial defect density ($2 \times 10^{11}$ cm$^{-2}$) than

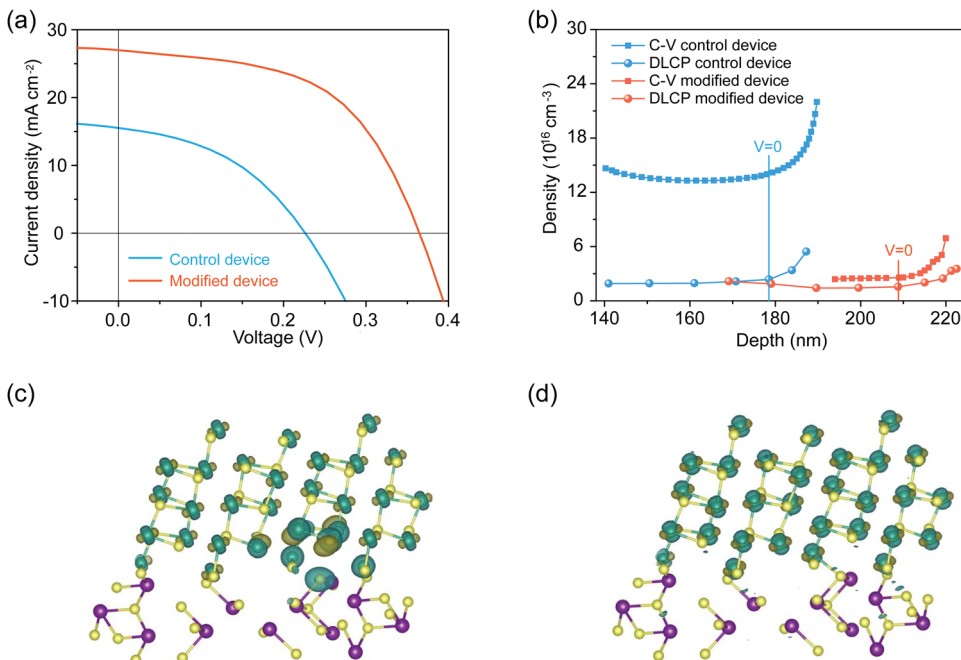

**Fig. 2 Analysis of device performance. a** J-V curves of control and modified GeSe devices. **b** C-V and DLCP characteristics of control and modified GeSe devices. DFT models for **c** trap like localized defects on the surface of GeSe film and **d** delocalized surface defects on GeSe after passivation.

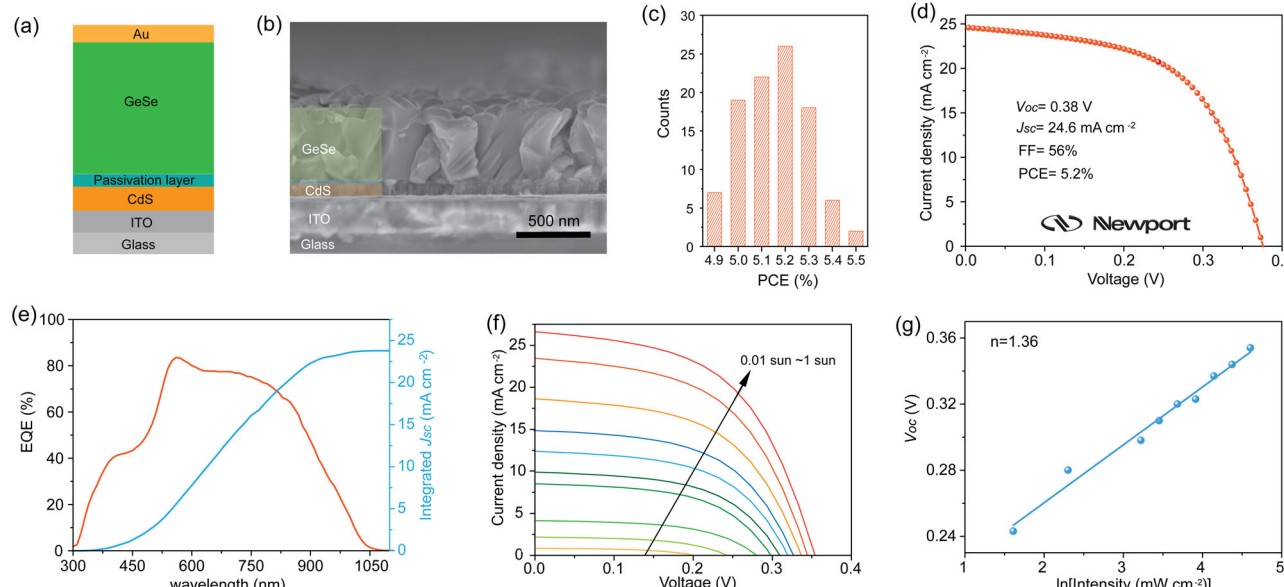

**Fig. 3 Photovoltaic performance. a** Schematic of GeSe thin-film solar cell architecture. **b** Cross-sectional SEM image of the GeSe device. **c** Histogram of device efficiencies obtained from 100 devices. **d** J-V curve and **e** EQE spectrum of the GeSe solar cell independently certified by Newport Corporation (Newport Corporation PV Laboratory, certificate #1896). **f** J-V curves of a representative GeSe device measured under different intensities of simulated AM 1.5 G illumination. **g** Light intensity-dependent $V_{oc}$ of GeSe solar cells. Neutral-density filters (THORLABS) were used to adjust the light intensity.

in the control devices ($2 \times 10^{12}$ cm$^{-2}$) (Fig. 2b). Achieving high-performance GeSe solar cells will require further work on the passivation of surface defects rather than bulk defects.

We fabricated over 100 GeSe solar cells (device architecture in Fig. 3a). Figure 3b shows a cross-sectional scanning electron microscope (SEM) image of a device; mapping with false coloring delineates the layers. The thickness of the GeSe layer is 500 nm, and the thickness of the passivation layer is 10 nm (Supplementary Fig. 8). The average grain size of the GeSe film is 250 nm (Supplementary Fig. 9). There is a narrow distribution of PCE values (Fig. 3c), with an average efficiency of 5.2% and a standard

deviation of 0.14%. The best-performing device reaches a laboratory PCE of 5.5% ($V_{oc} = 0.36$ V, $J_{sc} = 26.6$ mA cm$^{-2}$, and FF = 57%) (Supplementary Fig. 10). No hysteresis is observed between forward and reverse scans.

We shipped an unencapsulated device to an accredited independent photovoltaic testing laboratory (Newport Corporation PV Lab, USA). This device displays a certified PCE of 5.2% (Fig. 3d), with a corresponding $V_{oc}$ of 0.38 V, $J_{sc}$ of 24.6 mA cm$^{-2}$, and FF of 56% (accreditation certificate in Supplementary Fig. 11). This is the highest PCE reported so far for GeSe solar cells. Integration of the external quantum efficiency (EQE) collected

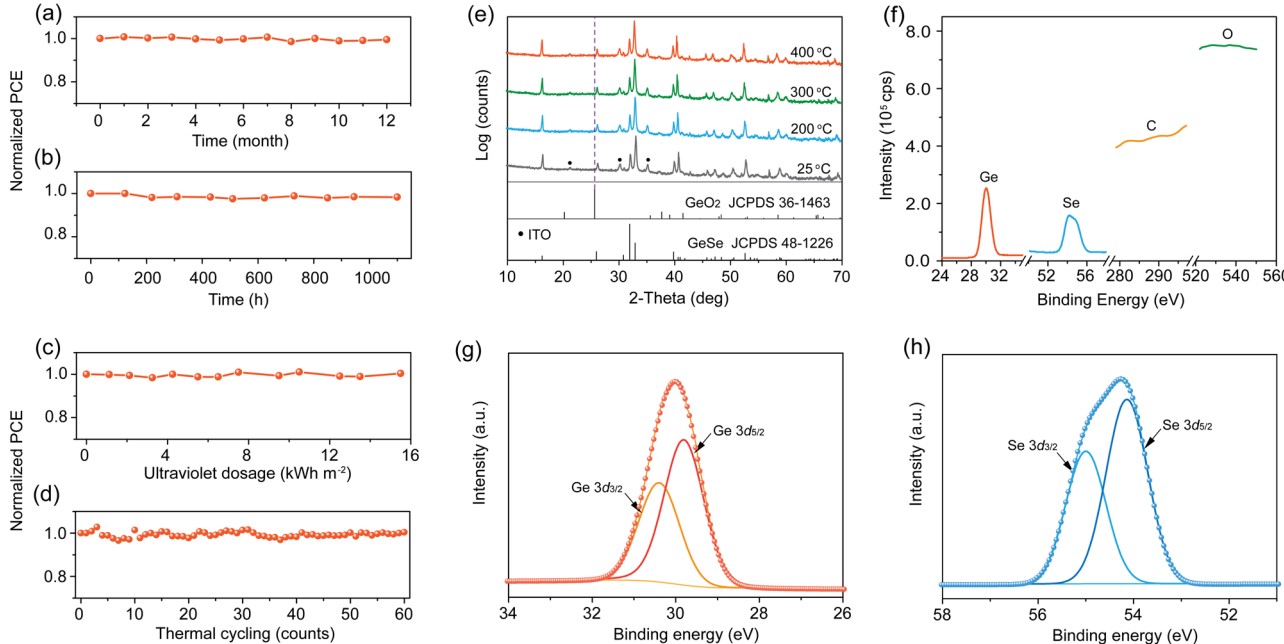

**Fig. 4 Device and materials stability. a** Long-term stability (ambient atmosphere, room temperature, relative humidity of 50–85%), **b** operational stability (ambient atmosphere, continuous 1-sun illumination, close to maximum power point), **c** ultraviolet photostability (200–400 nm ultraviolet light irradiation), and **d** thermal cycling stability (cycling between −40 to 85 °C for 60 cycles) of unencapsulated GeSe devices. **e** Temperature-dependent XRD patterns of GeSe film from 25 to 400 °C in ambient atmosphere. XPS spectra of **f** Ge, Se, C, and O, **g** Ge 3d, and **h** Se 3d in the GeSe film after temperature-dependent XRD measurements.

under the standard AM 1.5 G solar spectrum yields a current density of 23.8 mA cm$^{-2}$ (Fig. 3e), in good agreement with the $J_{sc}$ value measured from $J$-$V$ characterization (within 5% deviation) and also consistent with the absorption edge of GeSe (Supplementary Fig. 12). When we measured device performance at low-light intensities (Fig. 3f), we found that devices exhibit PCE values of 5.3%, 6.3%, and 8.6% (Supplementary Table 1) as we progress down to 0.01 sun. The corresponding light intensity-dependent $J_{sc}$ and $V_{oc}$ are shown in Supplementary Fig. 13 and Fig. 3g. The power value α (0.85) obtained from fitting to $J_{sc}$ measurement is close to unity (first-order); the slope obtained by linear fitting from $V_{oc}$ measurement is 1.36($k_BT/q$), larger than $k_BT/q$ for trap-free solar cells, indicating that trap-assisted recombination is still present in these GeSe devices.

**Device and materials stability.** Stability of the GeSe devices was monitored by storing unencapsulated devices in an ambient atmosphere at room temperature and a relative humidity of 50–85%. Devices retain 100% of their initial PCE after storage for 12 months (Fig. 4a). They also show negligible efficiency loss after continuous operation close to the maximum power point (MPP) under 1-sun illumination for 1100 h (Fig. 4b). We then investigated the ultraviolet photostability of unencapsulated devices under ultraviolet irradiation (200–400 nm). They retain their efficiency after exposure to an ultraviolet irradiation dosage of 15.5 kWh m$^{-2}$ (Fig. 4c). The thermal stability was investigated by cycling the temperature from −40 to 85 °C for a total of 60 cycles. They show no loss of efficiency after 60 thermal cycles (Fig. 4d).

Temperature-dependent XRD under an ambient atmosphere was applied to explore the origin of air and thermal stability. The film keeps its orthorhombic GeSe (JCPDS 48-1226) phase with no impurity peaks observed (such as GeO$_2$) even up to 400 °C for 30 min (Fig. 4e). Since XRD is unable to detect amorphous components, we performed X-ray photoelectron spectroscopy (XPS) and energy-dispersive X-ray spectroscopy (EDS) on the same GeSe film after temperature-dependent XRD

measurements. Oxygen and carbon are not detected in GeSe films (see magnified XPS spectrum at 520–550 eV and 281–295 eV) (Fig. 4f), consistent with the EDS results (Supplementary Fig. 14). The Ge 3d$_{5/2}$ and Ge 3d$_{3/2}$ peaks in the Ge 3d spectrum are observed at 29.85 eV and 30.43 eV (Fig. 4g), corresponding to Ge in the +2 oxidation state[24]. Gaussian-Lorentzian fitting confirms that no peak corresponding to +4 or 0 state of Ge is observed within the detection limit of the XPS instrument. The Se 3d spectrum also reveals that Se is in the expected oxidation state of Se$^{2-}$, corresponding to GeSe (Fig. 4f)[30]. The above results, therefore, demonstrate the high air and thermal stability of GeSe. In addition, GeSe also exhibits excellent humidity and light stability (Supplementary Fig. 15). The 4s$^2$ electrons on the Ge cation in ionic perovskites are exposed, making them vulnerable to oxidation; while the lone-pair electrons on the Ge atoms in covalent GeSe participate in Ge 4s–Se 4p coupling, leading to chemical inactivity.

## Discussion
In summary, we report a binary and non-toxic photovoltaic absorber material, GeSe, with benign defect properties and high stability arising from its antibonding VBM from Ge 4s–Se 4p coupling. We found photovoltaic devices to be limited by surface defects rather than by bulk defects. By passivating these interfacial defects, we achieved a certified record PCE of 5.2%. Unencapsulated GeSe devices exhibited no performance degradation under long-term ambient air, operating, ultraviolet soaking, and thermal cycling conditions. This work provides a deep understanding of the relationship between chemical bonding and macroscopic, device-relevant properties including the nature of defects and material's stability.

## Methods
**Solar cell fabrication.** All devices were deposited on ITO (Sn-doped In$_2$O$_3$) conductive glass, which was cleaned using detergent, deionized water, acetone, and isopropanol in sequence. GeSe thin-film solar cells were fabricated with a structure

consisting of CdS, passivated layer, GeSe, and Au. First, a CdS buffer layer was deposited by chemical bath deposition (CBD) on ITO conductive glass according to a previous report[24]. Then, the passivation layer ($Sb_2Se_3$) was deposited using a modified rapid thermal evaporation (RTE) method[40]: a tube furnace was set at 300 °C for 15 min to warm up the substrate, before raising the temperature to 550 °C in 30 s to start the evaporation. We then kept to this temperature and carried out 2 s of deposition to obtain the 10 nm thick $Sb_2Se_3$ layer, then the power is turned off to stop the evaporation, and finally, the film was removed when it was cooled to 180 °C. We then transferred this $Sb_2Se_3$ modified substrate to another tube furnace in ambient air without protection; this next sept was to enable the deposition of GeSe film. GeSe films were fabricated by rapid thermal sublimation method as in a previous report[24]: we preheated the GeSe powder and substrate at 350 °C for 20 min, before quickly increasing the source temperature to 400 °C within 2 s, maintaining this temperature for 5 s, before finally turning off the heating. Finally, Au back contacts (80 nm) were deposited using a thermal evaporation system (Beijing Technol Science) through a shadow mask (0.09 $cm^2$).

**Materials characterization**. Powder XRD patterns were recorded using a Rigaku D/Max-2500 diffractometer with a Cu target (Kα1 radiation, $\lambda = 1.54056$ Å). High-resolution XPS measurements were performed on an ESCALab220i-XL electron spectrometer (VG Scientific) using 300 W Al Kα radiation. The optical transmittance was measured using a UV − vis-near IR spectrophotometer (UH4150, HITACHI). Scanning electron microscopy (SEM) cross-sectional images were obtained by Hitachi S-4800 microscope. Atomic force microscopy (AFM) data were collected on a Bruker Dimension Icon microscope.

**Device performance characterization**. J-V curves of the solar cells were obtained using an AM 1.5 G solar simulator (Newport, USA) equipped with a Keithley 2420 source meter and 450 W xenon lamp (OSRAM) in the air at room temperature. Light intensity was adjusted using an NREL certified Si solar cell with a KG − 2 filter for approximating AM 1.5 G light (100 mW $cm^{-2}$). The device was covered with a metal mask with an aperture area of 0.09 $cm^2$ during efficiency measurement. The J-V curves were measured with a scanning rate of 100 mV $s^{-1}$ (voltage step of 20 mV and delay time of 200 ms). Both forward (−1 to 1 V) and backward (1 to −1 V) scans were recorded.

**Electrical characterization**. DLTS measurements were performed using an FT-1030 HERA DLTS system equipped with a JANIS VPF-800 cryostat controller on the high-performance device. The temperature was scanned between 140 and 380 K. The reverse bias voltage was set to −0.5 V. The filling pulse voltage and width were 0.4 V and 20 ms, respectively. The C-V curves and DLCP spectra were obtained using a Keithley 4200. C-V measurements were performed at room temperature in an electromagnetic shielding box at a frequency of 10 kHz and an A.C. amplitude of 30 mV. The D.C. bias voltage was scanned from −1.0 V to 0.5 V with a step size of 0.01 V. DLCP measurements were performed with an A.C. amplitude ranging from 14 mV to 140 V and D.C. bias voltage from −0.2 V to 0.2 V.

**Light-soaking test**. Devices were illuminated using a Xe light source (PLS-SXE300, 1.35 sun intensity) with an AM 1.5 light filter. A 160 Ω resistor was connected and the load was measured to be 175 Ω due to the contribution from the additional connection circuit. The device was continuously operated near the MPP, and the measured output current density was ~32 mA $cm^{-2}$. The temperature of the device was maintained at 45–55 °C without any external cooling. Every 100 h, the device was subjected to the standard AM 1.5 to collect J-V curves as described above after cooling down to room temperature within a few minutes.

**Ultraviolet stability tests**. Devices were exposed to UV light with wavelength range from 200 to 400 nm (Xe light source, PLS-SXE300) for 62 h and the power was kept at 25 mW $cm^{-2}$. The devices were put on a hot plate maintaining the temperature at 60 °C. After illumination, we took out the device and measured the J-V curves under AM 1.5 illumination. Please note that compared to the IEC 61646 protocol, our measurement included a deeper range of ultraviolet light (for the 200–280 nm components).

**Thermal cycling test**. The devices were placed inside a high-low temperature test chamber under the atmosphere. Chamber temperature was periodically changed from −40 to 85 °C for 60 cycles. For every cycle, the total time is 70 min and the device temperature remained stable at each extreme for 10 min. After each cycle, the device temperature was adjusted to 298 K and device performance was measured.

**Density functional theory**. The crystal structure, total energy, and bandstructure were calculated using DFT methods as implemented in the Vienna ab initio simulation package (VASP) code[41]. Frozen-core projector augmented-wave (PAW) pseudopotentials and a plane wave basis set with an energy cutoff of 520 eV were employed, with a $8 \times 8 \times 4$ Monkhorst-Pack **k**-point mesh included in the Brillouin zone integration for the 16-atom primitive cell and a $3 \times 3 \times 3$ mesh for the 64-

atom supercell, which was used for the calculation of defect properties[32]. Test calculations were performed with a higher energy cutoff, denser **k**-point mesh, and larger supercell size, and we found the same trends. The GGA to the exchange-correlation functional was used in all the calculations, which predicted a bandgap around 0.9 eV for GeSe, slightly lower than the experimental value at 1.1 eV. To estimate the influence of GGA on the calculated results, the non-local hybrid functional was also used which predicted the bandgap more accurately[42], showing that the conclusions were not influenced by the specific functionals.

**Reporting Summary**. Further information on research design is available in the Nature Research Reporting Summary linked to this article.

## Data availability

The data that support the findings of this study are available on reasonable request from the corresponding author.

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

## Acknowledgements

This work is supported by the National Natural Science Foundation of China (21922512, 21875264, 61725401), the Youth Innovation Promotion Association CAS (2017050). The work of Y.M., Y.H., A.P., and E.H.S. is supported by the US Department of the Navy, Office of Naval Research (Grant Award NO. N00014-17-1-2524).

## Author contributions

D.-J.X. conceived the idea and designed the experiments. S.-C.L. prepared films, fabricated devices, and characterized them. C.-M.D., Y.M., and S.C. performed the DFT calculations and analyzed the results. Y.Z. and C.C. assisted in the device characterization. Y.H. and A.P. helped with the manuscript preparation. J.T. discussed the results and commented on the paper. D.-J.X., S.-C.L., and E.H.S. wrote the paper. E.H.S. supervised the manuscript preparation. D.-J.X. and J.-S H. supervised the project. All authors read and commented on the manuscript.

## Competing interests

The authors declare no competing interests.
