## [Peer Review File · Nature Communications]

REVIEWER COMMENTS

Reviewer #1 (Remarks to the Author):

This paper reports a fourfold increase in GeSe solar cell device efficiency from the previous current record of 1.48% to 5.2%. If this was a reasonable representation of the findings, then this would be sufficiently exciting and interesting to justify publication in Nature Communications.

However, and this is a big however, the significant increase in power conversion efficiency results from the use of a "passivation" between the GeSe and CdS. This passivation layer is Sb₂Se₃. I was unable to find the thickness of this layer - it is referred to as being deposited to the "desired thickness". CdS/Sb₂Se₃ junctions are well known to produce reasonable solar cells, with very many reports of efficiencies over 5% and the record of 9.2%. So I am suggesting that what is being reported may be a CdS/Sb₂Se₃ solar cell with an additional GeSe layer. If this is the case, then this is not so exciting and publication in Nature Communications would not be appropriate.

So how thick is the Sb₂Se₃ layer? And what evidence do you have for its thickness?

Can you properly index the XRD peaks of the "modified" GeSe layer - are any due to Sb₂Se₃?

You state a band gap of GeSe of 1.14 eV, but give no evidence for that. Your DFT gap is 0.9 eV, but, as you say, this will underestimate the gap as it is a GGA calculation. Can you provide optical data for the band gap of GeSe? A recent report from March 2020 indicates that it is about 1.3 eV (doi.org/10.1021/acs.chemmater.0c00453).

This leads to questions about the EQE - the band gap of Sb₂Se₃ is about 1.18 eV, so are some features of the EQE due to that rather than GeSe? Has a reasonable solar cell of Sb₂Se₃/CdS been made with a thinnish layer of Sb₂Se₃ but then with GeSe on the back to enhance absorption at shorter wavelength?

So I suggest either:

- i) the paper be reworked in terms of the role of the Sb₂Se₃ layer including referring to the solar cell with 5.2% efficiency as a CdS/Sb₂Se₃/GeSe device and then submit to a different journal; or
- ii) address all of the points with evidence of the thinness of the passivation layer and optical characterization of the GeSe and a full justification of all the EQE and arguments as to why my interpretation is not correct - then Nature Comms might be possible.

Reviewer #2 (Remarks to the Author):

The paper reports on a GeSe solar cell with a power conversion efficiency (PEC) of around 5%. Based on density functional theory (DFT) calculations the authors propose that the main reason for the "high" efficiency (compared to earlier reports on the the same material which have shown PEC of 1-2%) is that the surface defects in the current work were passivated by chemical saturation with Sb₂Se₃.

I have several reservations with this manuscript and cannot recommend publication in Nature Communications. Most importantly, it is unclear what the novelty of the work is. The concept of defect tolerance is not new and has been presented before in several contexts and materials. The fact that GeSe should be defect tolerant is not very surprising nor interesting, although it is of course relevant in the context of using GeSe as photoabsorber. The importance of the paper does not lie in the material either as GeSe is a well known semiconductor. While a PEC of 5% is maybe a record for GeSe, it is not very impressive. Therefore, while the paper is interesting to the PV community, it is not novel enough to warrant publication in a high-profile journal like Nature Com.

More specific questions:

- 1) The authors state that GeSe is defect tolerant because its VBM has anti-bonding character. But for a truly defect tolerant material, the CBM should also be anti-bonding. What is the situation for GeSe?
- 2) The PBE xc-functional is not reliable for defect energy levels (charge transition energies). Instead the HSE functional is the gold standard. The authors should show for at least one of the considered defects that their PBE results (regarding the classification of shallow/deep level) are reliable.
- 3) The V_{oc} even after surface passivation, is only 0.36 V while the band gap is 1.1 eV. The large difference indicates that defects play a significant role as recombination centers contrary to what claimed. How do the authors explain the low V_{oc} if it is not related to defects?
- 4) The device contains CdS next to the GeSe film. CdS is a well known photovoltaic material. How can the authors rule out that (part of) the measured PEC is due to the CdS rather than GeSe?

Reviewer #3 (Remarks to the Author):

The authors of the paper "An antibonding valence band maximum enables 2 defect-tolerant and stable GeSe photovoltaics" report on a GeSe PV structure with 5.2% efficiency which is higher than previously reported. This was accomplished by passivating interfacial defects. The efficiency is impressive and the paper also includes some informative characterization and it is suggested that the paper could be published after improving the following:

The strength of the paper is in the characterization of the unilluminated system and the characterization of defects and the electronic structure of the valence band including the VBM. However, for PV functionality the properties of the CB are as important and the discussion of the frontier electronic structure should generally be improved in this part. Examples include.

- a) Extend the Fig 1c including the energy levels/hybridization in the conduction band.
- b) What is the character of the conduction band. Effects from Se 4s?

It is understood that controlling interface defects are important for improved PV functionality. Elaborate on the strategy. "We devise therefore a surface passivation strategy...". What was the strategy? It is unclear to what extent the Sb₂Se₃ deposition was optimized in terms of thickness and other parameters (e.g. temperature). How did for example thickness affect interfacial defect density. It is also unclear how the substrate was transported between depositions of the different materials (vacuum, presence of oxygen/water?). The interface design is central in the conclusions and the experimental description should be clarified.

Linked to the above XPS is a valuable technique in characterizing surfaces/interfaces and measurements of the different surfaces (before deposition of the next layer) could give valuable information. Liked to this it is unclear how "clean" the surface is from oxygen (and carbon not shown). This data should not be shown with a logarithmic y-axis. XPS spectra such as those in Fig 4 g,h for O1s (and C1s) should be included and quantified (estimation) using sensitivity factors.

The characterization of illuminated devices is very limited. One possibility to do this would be to include the Supplementary Figure 9 to the main manuscript and interpret these. Moreover, it is more common to show the photovoltage versus light intensity on a logarithmic scale than photocurrent versus light intensity (as is the case here).

There are some minor typos (also in the abstract) that should be corrected (e.g. bond should be band).

Response Letter to Reviewers' Comments

For Reviewer #1 :

This paper reports a fourfold increase in GeSe solar cell device efficiency from the previous current record of 1.48% to 5.2%. If this was a reasonable representation of the findings, then this would be sufficiently exciting and interesting to justify publication in Nature Communications.

However, and this is a big however, the significant increase in power conversion efficiency results from the use of a "passivation" between the GeSe and CdS. This passivation layer is Sb_2Se_3 . I was unable to find the thickness of this layer - it is referred to as being deposited to the "desired thickness". CdS/ Sb_2Se_3 junctions are well known to produce reasonable solar cells, with very many reports of efficiencies over 5% and the record of 9.2%. So I am suggesting that what is being reported may be a CdS/ Sb_2Se_3 solar cell with an additional GeSe layer. If this is the case, then this is not so exciting and publication in Nature Communications would not be appropriate.

Comment 1: So how thick is the Sb_2Se_3 layer? And what evidence do you have for its thickness?

Response 1: To characterize the thickness of Sb_2Se_3 layer, we carried out cross-sectional scanning electron microscopy (SEM), atomic force microscopy (AFM), and transmission spectroscopy. As shown in Fig. 3b and the newly added Supplementary Fig. 8a, SEM and AFM characterization shows that the Sb_2Se_3 layer is about 10 nm thick. A comparison of transmittance spectra of pure CdS and CdS/ Sb_2Se_3 films also attests to the fact that the Sb_2Se_3 layer is indeed thin: there is no obvious change of transmittance spectra when moves from CdS to CdS/ Sb_2Se_3 films (Supplementary Fig. 6c). Considering the narrow bandgap of Sb_2Se_3 (~1.18 eV direct) with high optical absorption coefficient ($> 10^5 \text{ cm}^{-1}$) compared with that of CdS (2.4 eV), this also indicates that the Sb_2Se_3 is $< \sim 10 \text{ nm}$. In the revised manuscript, we have added Supplementary Fig. 8 and relevant discussion to make this point clear.

Supplementary Figure 8. (a) Cross-sectional SEM image of the GeSe solar cell. (b) AFM image of Sb_2Se_3 layer.

The thickness of the GeSe layer is 500 nm; and the thickness of the passivation layer is 10 nm (Supplementary Fig. 8). (Page 10)

Comment 2: Can you properly index the XRD peaks of the "modified" GeSe layer - are any due to Sb_2Se_3 ?

Response 2: As shown in the updated Supplementary Fig. 7b, the main XRD peaks of modified GeSe layer are indexed to the orthorhombic GeSe (JCPDS 48-1226), and the peaks labelled with "•" belong to ITO, whereas it is not clear for the XRD peaks of Sb_2Se_3 . This may be attributed to the different thickness of functional layers. The thicknesses of GeSe and ITO layers are about 500 nm and 200 nm, whereas the Sb_2Se_3 layer is as thin as 10 nm, as discussed above.

Supplementary Figure 7. (b) XRD patterns of control and modified GeSe films.

Comment 3: You state a band gap of GeSe of 1.14 eV, but give no evidence for that. Your DFT gap is 0.9 eV, but, as you say, this will underestimate the gap as it is a GGA calculation. Can you provide optical data for the band gap of GeSe? A recent report from March 2020 indicates that it is about 1.3 eV (doi.org/10.1021/acs.chemmater.0c00453). This leads to questions about the EQE - the band gap of Sb_2Se_3 is about 1.18 eV, so are some features of the EQE due to that rather than GeSe? Has a reasonable solar cell of $\text{Sb}_2\text{Se}_3/\text{CdS}$ been made with a thinnish layer of Sb_2Se_3 but then with GeSe on the back to enhance absorption at shorter wavelength?

Response 3: We now provide transmission spectroscopy to characterize the bandgap of as-prepared GeSe films, which were fabricated using the RTS method. As shown in the newly-added Supplementary Fig. 12a, the transmittance declines at wavelength ~ 1100 nm. We further used Tauc plot method to fit the bandgap of GeSe film, and obtained a bandgap of about 1.14 eV (Supplementary Fig. 12), agreeing well with previously reported values (*J. Am. Chem. Soc.* 2010, 132, 43, 15170; *J. Am. Chem. Soc.* 2017, 139, 2, 958). We have added Supplementary Fig. 12 into the revised manuscript.

We also better discuss the EQE, integrated J_{sc} , as they relate to the measured bandgap of GeSe (1.14 eV). In this discussion we note that the Sb_2Se_3 layer is thin enough (~ 10 nm) that it contributes negligibly to absorption.

Supplementary Figure 12. (a) Transmittance spectrum of GeSe film prepared by rapid thermal sublimation (RTS) method; (b) Tauc plot ($n = 1/2$, indirect) for GeSe film.

Integration of the external quantum efficiency (EQE) curve collected under a standard AM1.5G solar spectrum yields a current density of 23.8 mA cm^{-2} (Fig. 3e), in good agreement with the J_{sc} value measured from our J-V characterization (within 5% deviation) and also consistent with the absorption edge of GeSe (Supplementary Fig. 12). (Page 10)

Comment 4: So I suggest either: i) the paper be reworked in terms of the role of the Sb_2Se_3 layer including referring to the solar cell with 5.2% efficiency as a CdS/ Sb_2Se_3 /GeSe device and then submit to a different journal; or ii) address all of the points with evidence of the thinness of the passivation layer and optical characterization of the GeSe and a full justification of all the EQE and arguments as to why my interpretation is not correct - then Nature Comms might be possible.

Response 4: We now provide evidence of the thin Sb_2Se_3 layer (~ 10 nm) and of the smaller GeSe bandgap (~ 1.14 eV) than Sb_2Se_3 (1.18 eV) and discuss the observed EQE values in this context. We have added supplementary Fig. 8 and 12 and relevant discussions to make these points clear.

For reviewer #2:

The paper reports on a GeSe solar cell with a power conversion efficiency (PEC) of around 5%. Based on density functional theory (DFT) calculations the authors propose that the main reason for the "high" efficiency (compared to earlier reports on the the same material which have shown PEC of 1-2%) is that the surface defects in the current work were passivated by chemical saturation with Sb_2Se_3 .

Comment: I have several reservations with this manuscript and cannot recommend publication in Nature Communications. Most importantly, it is unclear what the novelty of the work is. The concept of defect tolerance is not new and has been presented before in several contexts and materials. The fact that GeSe should be defect tolerant is not very surprising nor interesting, although it is of course relevant in the context of using GeSe as photoabsorber. The importance of the paper does not lie in the material either as GeSe is a well known semiconductor. While a PEC of 5% is maybe a record for GeSe, it is not very impressive. Therefore, while the paper is interesting to the PV community, it is not novel enough to warrant publication in a high-profile journal like Nature Com.

Response: In the revised work we now more clearly discuss the concept of defect tolerance, including reports on this topic in prior works, including on perovskite. We make it more clear that defect tolerance has not been reported in GeSe until the present manuscript. We point to the use of both deep-level transient spectroscopy (DLTS) and admittance spectroscopy (AS) measurements which directly identify, experimentally, the low density of deep level defects in GeSe (about 10^{12} cm^{-3}).

We also now more clearly discuss the oxidation vulnerability of Ge-perovskites from Ge^{+2} to Ge^{+4} ; and how the present work contrasts with these prior reports, revealing strikingly high stability of GeSe. We find that GeSe keeps its orthorhombic phase with no GeO_2 even under high temperature 400°C in air.

We also discuss the fact that GeSe solar cells were first reported in 2017. As reviewers 1 and 3 note, moving in ~ 3 years from 1.5% to 5.2% represents good progress for this heavy-metal-free active layer.

More specific questions:

Comment 1: The authors state that GeSe is defect tolerant because its VBM has anti-bonding character. But for a truly defect tolerant material, the CBM should also be anti-bonding. What is the situation for GeSe?

Response 1: We now more clearly discuss on page 5 that the CBM of GeSe also has an antibonding character and provide support in the form of Supplementary Fig. 3.

Supplementary Figure 3. (a) Electronic structure of defect-intolerant semiconductors. (b) Electronic structure of defect-tolerant semiconductors for both acceptor and donor defects. (c) Electronic structure of defect-tolerant semiconductors for acceptor defects. (d) Electronic structure of defect-tolerant semiconductors for donor defects.

The CBM also has an antibonding character originating from the Ge 4p-Se 4p coupling. (Page 5)

The shallow nature of V_{Ge} originates from the antibonding state at the VBM, a defect-tolerant electronic structure known to lead to shallow defects (Supplementary Fig. 3). (Page 6)

Comment 2: The PBE xc-functional is not reliable for defect energy levels (charge transition energies). Instead the HSE functional is the gold standard. The authors should show for at least one of the considered defects that their PBE results (regarding the classification of shallow/deep level) are reliable.

Response 2: We now report use of the HSE functional to investigate the defect energy levels of GeSe. The values of formation energies and transition energy levels are substantially unchanged relative to PBE results. In the revised manuscript, we have added the HSE functional results and relevant discussion to make this point clear.

Supplementary Figure 2. Calculated formation energies of intrinsic point defects used HSE function in GeSe under (a) Ge-rich and (b) Se-rich conditions as a function of the Fermi energy.

The second notable feature is that V_{Ge} , with the lowest formation energy, has a shallow level with (-/0) and (2-/-) transition energy levels only 0.05 and 0.15 eV above the VBM, whereas defects with deep levels such as Ge_{Se} , Ge_i , V_{Se} and Se_{Ge} have high formation energies (Supplementary Fig. 1). These are reconfirmed by Heyd-Scuseria-Ernzerhof (HSE) calculation (Supplementary Fig. 2). (Page 5)

Comment 3: The V_{oc} even after surface passivation, is only 0.36 V while the band gap is 1.1 eV. The large difference indicates that defects play a significant role as recombination centers contrary to what claimed. How do the authors explain the low V_{oc} if it is not related to defects?

Response 3: We now discuss the V_{oc} of the devices, and opportunities to improve it: we discuss unfavorable band alignment between the p-absorber material and n-buffer layer; and the role of interfacial defects at the p-n junction interface. We now provide light-intensity-dependent J_{sc} and V_{oc} studies that allow us to discuss the opportunity to decrease trap-assisted recombination further going forward.

The corresponding light intensity-dependent J_{sc} and V_{oc} are shown in Supplementary Fig. 13 and Fig. 3g. The power value α (0.85) obtained from fitting to J_{sc} measurement is close to unity (first-order); the slope obtained by linear fitting from V_{oc} measurement is $1.36(kBT/q)$, larger than kBT/q for trap-free solar cells, indicating that trap-assisted recombination is still present in these GeSe devices. (Page 10)

Comment 4: The device contains CdS next to the GeSe film. CdS is a well-known photovoltaic material. How can the authors rule out that (part of) the measured PEC is due to the CdS rather than GeSe?

Response 4: We now more clearly discuss the fact that the EQE (Fig. 3e) diminishes at wavelengths shorter than CdS's absorption onset, a result of absorption and recombination in the CdS layer. In sum, CdS is not a net contributor to J_{sc} .

Fig. R1 Schematic band diagram of different layers of the GeSe solar cells.

For reviewer #3:

The authors of the paper “An antibonding valence band maximum enables defect-tolerant and stable GeSe photovoltaics” report on a GeSe PV structure with 5.2% efficiency which is higher than previously reported. This was accomplished by passivating interfacial defects. The efficiency is impressive and the paper also include some informative characterization and it is suggested that the paper could be published after improving the following:

Comment 1: The strength of the paper is in the characterization of the unilluminated system and the characterization of defects and the electronic structure of the valence band including the VBM. However, for PV functionality the properties of the CB are as important and the discussion of the frontier electronic structure should generally be improved in this part. Examples include: a) Extend the Fig 1c including the energy levels/hybridization in the conduction band. b) What is the character the conduction band. Effects from Se 4s?

Response 1: a) From the density of states (DOS) and partial DOS of GeSe shown in Fig. 1b, the conduction band of GeSe is composed of Ge 4p orbital and Se 4p orbital. This originates from the antibonding state of Ge 4p-Se 4p coupling. Accordingly, the updated Fig. 1c demonstrates hybridization in the conduction band of GeSe. b) From the partial DOS of GeSe projected on Se 4s orbital (Fig. 1b), there is little Se 4s character in the conduction band of GeSe. The conduction band of GeSe consists of Ge 4p and Se 4p states. In the revised manuscript, we have modified Fig. 1c and added discussion of the character of the conduction band of GeSe to make this point clear.

Fig. 1c Schematic energy level diagram of the interactions in GeSe

The conduction band minimum (CBM) of GeSe is dominated by the Ge 4p orbital, with significant coupling with the Se 4p orbital; and negligible coupling with the Se 4s orbital (Fig. 1b). (Page 5)

The CBM also has an antibonding character originating from the Ge 4p-Se 4p coupling. (Page 5)

Comment 2: It is understood that controlling interface defects are important for improved PV functionality. Elaborate on the strategy. “We devise therefore a surface passivation strategy...”. What was the strategy? It is unclear to what extent the Sb_2Se_3 deposition was optimized in terms of thickness and other parameters (e.g. temperature). How did for example thickness effect interfacial defect density. It is also unclear how the substrate was transported between depositions of the different materials (vacuum, presence of oxygen/water?). The interface design is central in the conclusions and the experimental description should be clarified.

Response 2: We now more clearly discuss the role of GeSe defect tolerance related to its ns^2 electronic configuration

We then more clearly motivate the surface passivation strategy, i.e. introducing a passivating Sb_2Se_3 layer to reduce interfacial defects between GeSe and CdS.

We then discuss the method of deposition of Sb_2Se_3 , citing Tang *et. al.* (*Nat. Photonics*, 2015, 409–415). That worked showed that the orientation of the Sb_2Se_3 layer is dependent on the substrate temperature, where low temperature leads to the [211]-oriented film, and high temperature results in the [120]-oriented layer. Our calculations showed that the growth of [111]-oriented GeSe on [221]-oriented Sb_2Se_3 is feasible while providing efficient passivation for the surface defects of GeSe (Fig. 2c and 2d).

We therefore adopted a low substrate temperature (300 °C), enabling us to deposit Sb₂Se₃ layer with the desired orientation of [221] (Supplementary Fig. 6a), and obtained [111]-oriented GeSe films (Supplementary Fig. 7).

We also improve the discussion of the fact that the layers were directly transported in air, without protection, testifying to the good air stability of Sb₂Se₃ and GeSe. Specifically, we first deposited a very thin Sb₂Se₃ layer on the substrate in a furnace tube and then took it out in ambient air. Next, we put the Sb₂Se₃-modified substrate in another tube and finish the deposition process of GeSe film.

In light of the reviewer's suggestions, we have added discussion of these points:

A tube furnace was set at 300°C for 15 min to warm up the substrate, before raising the temperature to 550°C in 30 s to start the evaporation. We then kept to this temperature and carried out 2s of deposition to obtain the 10 nm-thick Sb₂Se₃ layer, then the power is turned off to stop the evaporation, and finally the film was removed when it was cooled to 180°C. We then transferred this Sb₂Se₃ modified substrate to another tube furnace in ambient air without protection; this next sept was to enable the deposition of GeSe film.
(Page 13)

Comment 3: Linked to the above XPS is a valuable technique in characterizing surfaces/interfaces and measurements of the different surfaces (before deposition of the next layer) could give valuable information. Liked to this it is unclear how “clean” the surface is from oxygen (and carbon not shown). This data should not be shown with a logarithmic y-axis. XPS spectra such as those in Fig 4 g,h for O1s (and C1s) should be included and quantified (estimation) using sensitivity factors.

Response 3: We have changed the logarithmic y-axis of Fig. 4f to a linear one, and added C 1s in XPS spectra in Fig. 4f for comparison. From the updated Fig 4f, the signals associated with O1s and C1s are negligible, indicating the purity of the GeSe films, without appreciable oxygen and carbon impurities. The low O contamination is attributed to the high oxidation resistance of GeSe, while the low C contamination is due to the vacuum-based deposition method for GeSe film, where no carbon sources such as organic solvents are used in the film deposition process. We have updated Figure 4f and added relevant discussions to make these points clear.

Figure 4f. XPS spectra of (f) Ge, Se, C and O in the GeSe film after temperature-dependent XRD measurements.

Oxygen and carbon are not detected in GeSe films (see magnified XPS spectrum at 520-550 eV and 281-295 eV) (Fig. 4f). (Page 12)

Comment 4: The characterization of illuminated devices is very limited. One possibility to do this would be to include the Supplementary Figure 9 to the main manuscript and interpret these. Moreover, it is more common to show the photovoltage versus light intensity on a logarithmic scale than photocurrent versus light intensity (as is the case here).

Response 4: We now provide in the main manuscript in Fig. 3g the photovoltage vs. light intensity and added relevant discussion to make this point clear.

Fig. 3g Light intensity-dependent V_{oc} of GeSe solar cells.

The corresponding light intensity-dependent J_{sc} and V_{oc} are shown in Supplementary Fig. 13 and Fig. 3g. The power value α (0.85) obtained from fitting to J_{sc} measurement is close to unity (first-order); the slope obtained by linear fitting from V_{oc} measurement is $1.36(k_B T/q)$, larger than $k_B T/q$ for trap-free solar cells, indicating that trap-assisted recombination is still present in these GeSe devices. (Page 10)

Comment 5: There are some minor typos (also in the abstract) that should be corrected (e.g. bond should be band).

Response 5: We have reviewed and edited the manuscript.

REVIEWER COMMENTS

Reviewer #1 (Remarks to the Author):

The authors have address my main concern adequately. They now show evidence that the Sb₂Se₃ layer is 8-10 nm thick. So it really is a CdS/GeSe solar cell with a thin interlayer. I therefore recommend publication in Nature Comms after mandatory minor revision. The increase in efficiency of this technology is impressive.

I missed it in my first review, but the claim of a 4-fold increase is an exaggeration. 1.4% to 5.2% is not quite a four-fold increase - the authors should correct this so their claim is not over-stated.

Optical absorption data is now shown, in an attempt to justify the band gap value claimed. However, the Tauc plot shown is for an inidrect gap. GeSe may or may not be an indirect band gap material. However, if it is an indirect band gap material, the absorption and EQE curve will still be dominated by direct transitions, and so the appropriate Tauc plot should be shown, plotting $(\alpha \cdot h\nu)^2$ versus $h\nu$.

Reviewer #2 (Remarks to the Author):

I have read the authors response to mine and the other two referee reports. The authors have done a good job in responding carefully to criticism raised by the referees. In particular, they have provided further evidence in support of their conclusions. However, while the paper is interesting and deserves publication, I am still of the opinion that the main achievement of the paper can be categorised as device engineering while the scientific novelty is limited and not sufficient for Nature Communications.

Response Letter to Reviewers' Comments

Reviewer #1 :

The authors have addressed my main concern adequately. They now show evidence that the Sb_2Se_3 layer is 8-10 nm thick. So, it really is a CdS/GeSe solar cell with a thin interlayer. I therefore recommend publication in Nature Comms after mandatory minor revision. The increase in efficiency of this technology is impressive.

Comment 1: I missed it in my first review, but the claim of a 4-fold increase is an exaggeration. 1.4% to 5.2% is not quite a four-fold increase - the authors should correct this so their claim is not over-stated.

Response 1: We have corrected this claim from a 4-fold increase to a 3.7-fold increase to make it more accurate.

Comment 2: Optical absorption data is now shown, in an attempt to justify the band gap value claimed. However, the Tauc plot shown is for an indirect gap. GeSe may or may not be an indirect band gap material. However, if it is an indirect band gap material, the absorption and EQE curve will still be dominated by direct transitions, and so the appropriate Tauc plot should be shown, plotting $(\alpha h\nu)^2$ versus $h\nu$.

Response 2: We applied Tauc plot method based on direct transition— $(\alpha h\nu)^2$ versus $h\nu$ —to fit the bandgap of GeSe films. The direct bandgap of GeSe is 1.28 eV (Figure R1), matching previous reports. This direct bandgap is close to its indirect bandgap (1.14 eV), agreeing with previous results obtained from bandstructure calculations.

Figure R1. Tauc plot ($n = 2$, direct) for GeSe film.

Reviewer #2:

I have read the authors response to mine and the other two referee reports. The authors have done a good job in responding carefully to criticism raised by the referees. In particular, they have provided further evidence in support of their conclusions.

Comment: However, while the paper is interesting and deserves publication, I am still of the opinion that the main achievement of the paper can be categorised as device engineering while the scientific novelty is limited and not sufficient for Nature Communications.

Response: We have now clarified the following points in the revised work:

- (i) This is the first report of the defect-tolerant properties of GeSe. It is the first to link these properties to its antibonding VBM. We better explain the importance of contributing insights into defect-tolerant properties into the literature.
- (ii) We explore for the first time the unexpected oxidation resistance in GeSe; we better contrast it with the known oxidation-vulnerability of Ge-perovskites.